# Numerical analysis of sodium diffusion in aluminum electrolysis cathode carbon blocks based on a microstructure multi-factor corrected model

Chenglong Gong[1,2], Tianqi Xu [1,2,3]*, Huarong Qi[4], Yan Li[1,2,3]

**1** The Key Laboratory of Cyber-Physical Power System of Yunnan Universities, Yunnan Minzu University, Kunming, Yunnan Province, China, **2** School of Electrical and Information Technology, Yunnan Minzu University, Kunming, Yunnan Province, China, **3** Yunnan Key laboratory of Unmanned Autonomous System, Kunming, Yunnan Province, China, **4** School of Materials Science and Engineering, Kunming University of Science and Technology, Kunming, Yunnan Province, China

* xu.tianqi@ymu.edu.cn

**Data Availability Statement:** The content we submitted includes all the raw data required for replicating the research results, and we have shared our simulation data in the database as

## Abstract

Current researches on sodium penetration in electrolytic aluminum cathode carbon blocks primarily measure cathode expansion curves, showing mostly macroscopic characteristics. However, the microscopic structure is often underexplored. As a porous medium, the diffusion performance of cathode carbon blocks is closely tied to their internal pore structure. Viewing the cathode carbon block as a multiphase composite material, this study examines the sodium diffusion process from a microstructural perspective. A prediction model for sodium diffusion, considering factors like porosity, temperature, binding effects, current density, and molecular ratio, was developed. A random aggregate model was implemented in Python and imported into finite element software to simulate sodium diffusion using Fick's second law. Results indicate that increased porosity, higher temperatures, reduced binding effects, increased current density, and higher molecular ratios enhance sodium infiltration, reducing diffusion resistance and increasing the diffusion coefficient. The simulation aligns well with experimental results, confirming its accuracy and reliability.

## Introduction

Carbon cathodes are widely recognized as one of the pivotal components in contemporary aluminum electrolysis cells [1]. Throughout the electrolytic process, alkali metals such as sodium migrate from the molten material towards the cathode due to a chemical potential gradient, a phenomenon termed sodium permeation [2]. This infiltration behavior is profoundly detrimental, leading to carbon degradation, diminished current efficiency, and in severe cases, cell breakdown. Given the hostile conditions of high temperature and corrosive electrochemical environment, direct observation of permeation behavior proves exceedingly challenging. Consequently, the urgent imperative lies in simulating and thoroughly investigating the microstructure of cathode carbon blocks in practical settings.

required, as detailed in DOI:10.6084/m9.figshare.281327. If any additional data is needed, we are more than happy to provide it.

**Funding:** This work was financially supported by National Natural Science Foundation of China (62062068) and Yunnan Province Young and Middle aged Academic and Technical Leaders Reserve Talent Project (202305AC160077).

**Competing interests:** The authors have declared that no competing interests exist.

The mechanism of sodium permeation has been extensively studied, focusing on two primary perspectives: the steam migration mechanism [3] and diffusion within the carbon lattice [4]. Currently, it is widely accepted and experimentally confirmed that sodium migrates primarily through lattice diffusion. During electrolytic cell operation, sodium metal and high-temperature molten salt permeate the defect pores of the carbon block, causing corrosion. Sodium penetrates the carbon lattice layers, forming an embedded compound $Na_xC$ [5], which increases the interlayer spacing of the carbon lattice. This process reduces cohesion, internal friction angle, effective peak strength, and elastic modulus of the carbon block [6], progressively degrading its mechanical properties. Consequently, the carbon block's structural integrity undergoes expansion and cracking. The infiltration of sodium and electrolytes amplifies the cathodic pressure drop, with sodium penetration corrosion exacerbating the damage. Hence, assessing the diffusion behavior of sodium is critical for evaluating the performance of cathode carbon blocks.

Currently, research on sodium permeation predominantly revolves around measuring expansion curves of various cathodes, which are then fitted using mathematical or engineering approaches to elucidate the deformation law of permeation from a material mechanics perspective. However, these experiments are often time-consuming, labor-intensive, and tend to yield macroscopic characteristics, overlooking the microscopic structure of cathode carbon blocks. Simulation of cathode carbon blocks typically employs a homogeneous and isotropic model, with a limited focus on microscopic-scale sodium diffusion simulation. Wu [7] utilized homogenization theory to simulate sodium diffusion in cathode carbon blocks, treating them as a uniform model for research. Meanwhile, based on carbon block microstructures, Liu [8] considered cathode carbon blocks as a two-phase composite material comprising carbon aggregates and asphalt. They employed ANSYS finite element analysis software to establish a two-dimensional random aggregate model of cathode carbon blocks, investigating the impact of particle size composition, content, and morphology of carbon aggregates on sodium diffusion. However, these studies did not address environmental factors' influence on sodium diffusion or establish a sodium diffusion model that accurately reflects real-world conditions. Li [9] employed the Monte Carlo method and molecular dynamics simulations to generate a realistic atomic model of the anthracite cathode, investigating the internal mechanisms of sodium penetration. While this model provides detailed microscopic parameters, molecular modeling remains an immature field, and the behavior of atoms at the atomic scale may not fully correspond to their behavior at larger scales.

The carbon cathode material used in aluminum electrolysis cells typically exhibits a porosity ranging from 15 to 30 percent. These pores primarily result from initial microporous fissures within the aggregate that remain unfilled after the compression process, as well as from the arrangement of filling materials during fabrication and the calcined pores left behind after binder volatilization post-calcination [10]. Therefore, when constructing a microscopic model of cathode carbon blocks, the pore structure plays a critical role.

In this context, pore structure (original defects) is recognized as an integral phase of cathode carbon blocks, akin to a phase material. However, there remains a scarcity of studies in sodium diffusion simulation that comprehensively address the combined effects of multiple factors at the micro-scale. Hence, from a microscopic standpoint, it is especially crucial to delve deeper into and understand the sodium permeation behavior within cathode carbon blocks.

This paper aims to explore the influence of various factors on sodium diffusion within cathode carbon blocks through microscopic characterization. Initially, a two-dimensional random aggregate model of the cathode carbon block was developed using the Monte Carlo method. The model treats the cathode carbon block as a three-phase composite material comprising

carbon aggregates, asphalt, and pore structures (original defects). Subsequently, to apply the microscopic numerical simulation of sodium diffusion influenced by these factors to practical engineering, a sodium diffusion prediction model for cathode carbon blocks was formulated. Utilizing Fick's second law, the sodium diffusion process within the microstructure of cathode carbon blocks under different conditions was studied and analyzed. The prediction model was then validated against previous experimental results, demonstrating strong consistency with the empirical data. This validation underscores the accuracy and reliability of the simulation approach employed in this study.

## Establishment of computational models

### Microscopic random aggregate model for cathode carbon blocks

The random aggregate model has been extensively utilized in investigating chloride ion diffusion in concrete at the microscopic level, providing a theoretical foundation for the research presented in this paper. The random aggregate model is a method used for simulating and analyzing the microstructure of concrete. This model assumes that the composition of concrete (such as cement, aggregates, bubbles, etc.) is randomly distributed, allowing for the simulation of the effects of different types of aggregates on concrete performance. In concrete, the diffusion process of chloride ions is complex and influenced by factors such as pore structure, humidity, and temperature. The random aggregate model aids in the investigation of how chloride ions move between different aggregates and the cement matrix by simulating the microstructure of concrete. This model enables numerical simulations to generate extensive microstructural data, which can be compared with experimental results to validate the model's effectiveness. By adjusting model parameters, it is possible to predict the behavior and rate of chloride ion diffusion. In conclusion, the random aggregate model provides a robust framework for the in-depth study of chloride ion diffusion in concrete. Its theoretical foundation not only enhances the understanding of chloride ion diffusion phenomena but also offers significant references for subsequent research and applications. Despite the significant differences in the chemical properties of chloride ions and sodium, the similarities in diffusion mechanisms, physicochemical environments, material characteristics, and simulation methods at the microscopic level render their diffusion behaviors comparable to some extent. Consequently, these models can serve as effective tools in this study of sodium diffusion, aiding in the elucidation of issues related to sodium diffusion in cathode carbon blocks and promoting the advancement of research.

Considering the heterogeneous nature of the material, the cathode carbon block is conceptualized as a composite material consisting of carbon aggregate, asphalt binder, and pore structure. Previous research has demonstrated that the morphology of carbon aggregates significantly influences sodium diffusion capabilities [8]. To facilitate computational processes, the shape of the aggregate is conceptualized as a circle, and the pores are represented as circular voids. A novel algorithm for generating and placing random aggregates, which accounts for the gradation of carbon aggregates and pores, has been developed to establish the microstructure of cathode carbon blocks. The determination of carbon aggregate gradation involves two pivotal aspects: the particle size spectrum of the carbon aggregate and the proportion of each type of carbon aggregate with identical particle sizes. The ideal gradation curve of the mixing ratio can be described using the maximum density curve theory [11]. This theory can be expressed through the Fuller gradation formula as follows:

$$P = \sqrt{\frac{D}{D_{\max}}} \times 100\% \tag{1}$$

where $D$ is the particle size of a certain aggregate, $P$ is the percentage content of different aggregate particle sizes $D$, and $D_{max}$ is the maximum particle size of the aggregate in the grading group.

The two-dimensional mesoscale model, designed as a circle with a diameter of 50 mm, mirrors the cathode carbon block of the experimental sample. The carbon aggregate particle size ranges from 3 mm to 6 mm, and the pore size ranges from 0.5 mm to 0.8 mm. By adjusting the sample size, carbon aggregate fraction, porosity, aggregate gradation, and carbon aggregate pore size range, the desired micro geometric shape is achieved. The detailed steps are as follows:

1. Initial Setup:Determine the range for aggregate placement and calculate the total volume of aggregates and pores based on the input aggregate and pore volume fractions.

2. Generation of Aggregates and Pores: Using the grading curve and the Monte Carlo method, randomly generate aggregate and pore sizes within the specified particle size range. Generate different cathode carbon block aggregate content ratios based on the Fuller curve [11].

3. Volume Verification: Verify if the generated aggregate and pore volumes meet the predetermined total volume. Once they do, save them to the array.

4. Overlap and Range Check: Utilize the sequence of particle sizes from largest to smallest to determine if the aggregates and pores overlap or intersect. If there is any overlap or if the sizes exceed the range, the process is repeated until the requirements are met.

The algorithm process is depicted in Fig 1.

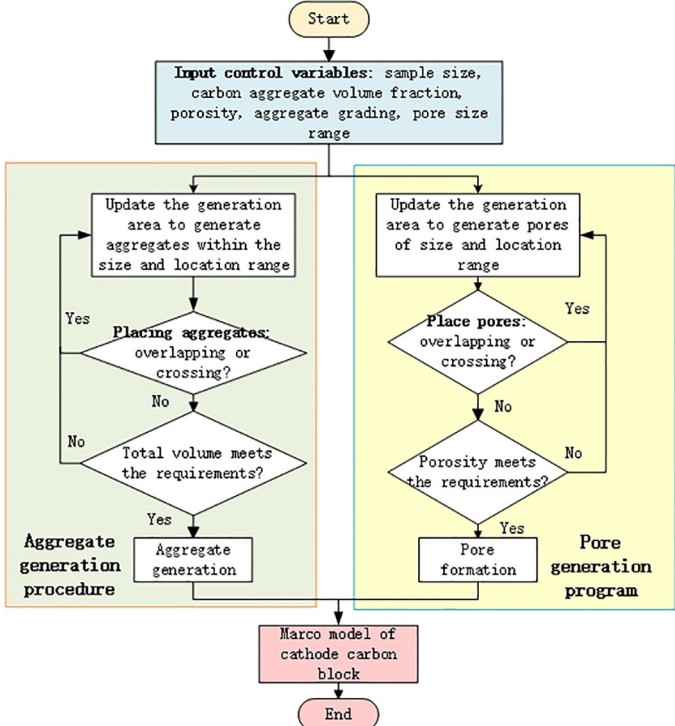

**Fig 1. Generation and filling algorithm of macro geometry of cathode carbon block.**

## Mathematical model of sodium diffusion

The transportation of sodium into the cathode carbon block involves various mechanisms, including diffusion, permeation, convection, capillary action, and electromigration, as well as their combined influences [8]. According to Lossius et al. [5], the penetration of sodium into carbon cathodes is primarily governed by diffusion, consistent with Fick's second law:

$$\frac{\partial C}{\partial t} = D \cdot \frac{\partial^2 C}{\partial x^2} \tag{2}$$

where $C$ is the sodium concentration, kg/m$^3$; $t$ is the exposure amount of the sample in the melt, s; $D$ is the sodium diffusion coefficient, m$^2$/s; $x$ is the depth of sodium penetration, m.

By solving the aforementioned equation and incorporating the boundary condition $C_s$ along with the initial concentration value $C_0$, a formula can be derived to determine the sodium concentration within the carbon cathode test block:

$$C(x, t) = C_0 + (C_s - C_0)\left[1 - erf\left(\frac{x}{2\sqrt{Dt}}\right)\right] \tag{3}$$

where $C_0$ is the surface concentration of sodium, kg/m$^3$; $erf$ is a Gaussian error function. The solution of the Gaussian error function in the equation is

$$erf\left(\frac{x}{2\sqrt{Dt}}\right) = \frac{2}{\sqrt{\pi}} \cdot \int_0^{\frac{x}{2\sqrt{Dt}}} e^{-y^2} dy \tag{4}$$

The sodium diffusion coefficient, denoted as $D$ in Eq (2), is conventionally considered as a constant value. However, in practical applications, sodium diffusion is characterized by its non-stationary nature and susceptibility to various micro and macroscopic factors, leading to fluctuations in the diffusion rate. Moreover, extrinsic environmental variables, such as temperature and current density, also influence the rate of sodium diffusion. Consequently, it is imperative to adjust the sodium diffusion coefficient to accurately depict the actual transmission process.

## Adjustment of sodium diffusion coefficient

Previous experimental findings have demonstrated a correlation between the deformation induced by sodium electrolyte infiltration and operational parameters, including the electrolyte's molecular ratio, the cathode's current density, and the electrolysis temperature. It has been observed that an increase in the molecular ratio, current density, and temperature leads to more pronounced infiltration of sodium electrolyte and, consequently, greater deformation. Therefore, it is imperative to investigate the sodium permeation behavior under the influence of these multifaceted factors from a microscopic perspective.

According to the experimental data presented in the study by Saitov et al. [12], the sodium diffusion coefficient increases with both temperature and duration of exposure. The effect of temperature can be quantitatively accounted for by the following correction formula:

$$f_T = \exp\left[\frac{E_0}{R}\left(\frac{1}{T_{\text{ref}}} - \frac{1}{T}\right)\right] \tag{5}$$

where $f_T$ is the temperature influence coefficient; $E_0$ is the ion activation energy, kJ/mol; $R$ is the molar gas constant, taken as 8.314 J/(mol · K); $T_{ref}$ is the reference temperature, taken as 1238.15 K; $T$ is the melt temperature, K. The variation pattern of $f_T$ with melt temperature is shown in Fig 2a.

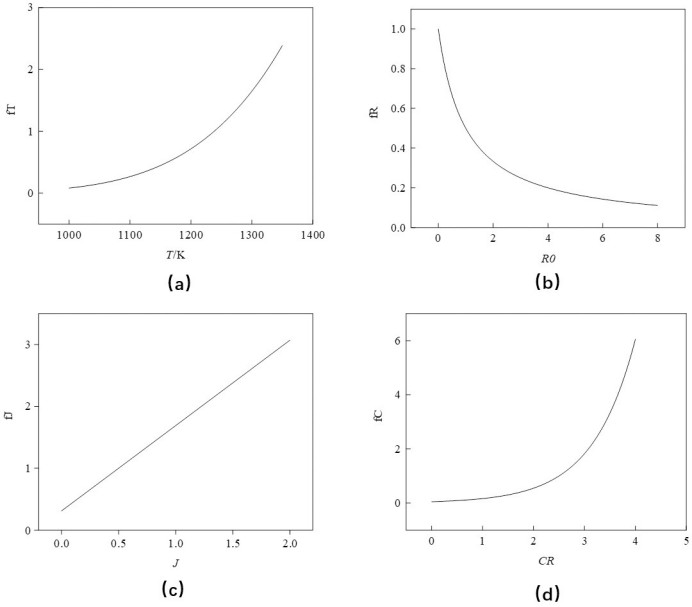

**Fig 2. Changes in various factors.** (**a**) Electrolysis temperature; (**b**) binding effect;(**c**) cathode current density; (**d**) melt molecular ratio.

A substantial body of analytical findings from planing furnaces suggests that approximately 23% of NaF is produced at the terminal end of the cathode and along the surface of the cathode steel rod [13]. This phenomenon is attributed to the formation of carbon-sodium compounds ($C_x$Na), which alter the wettability of the cathode carbon block. Consequently, the electrolyte infiltrates the cathode carbon block, initiating a series of reactions with sodium during the infiltration process:

$$Na_3AlF_6(l) + 3Na(c) = 6NaF(s) + Al(l) \tag{6}$$

If air enters the carbon lining, the following reactions may occur:

$$4Na_3AlF_6(l) + 12Na(c) + 3O_2 = 2Al_2O_3(s) + 24NaF(l) \tag{7}$$

$$4Na(c) + 3O_2 + 2C(s) = 2Na_2CO_3(s) \tag{8}$$

These two reactions are exothermic and tend to proceed significantly to the right, resulting in a substantial enrichment of sodium in the lower temperature zone adjacent to the cathode lining edge. Experimental results from Li et al. [10] indicate that sodium adsorption primarily occurs in larger micropores, ranging from 10 to 19 $\overset{\circ}{A}$, consistent with the Type I Langmuir adsorption model. By applying Fick's second law, the following solution can be derived:

$$\frac{\partial C_t}{\partial t} = \frac{\partial}{\partial x}\left(D \times \frac{\partial C_f}{\partial x}\right) \tag{9}$$

$$D_R = \frac{D}{1 + \frac{\partial C_b}{\partial C_f}} = \frac{D}{1 + R} \tag{10}$$

$$f_R = \frac{1}{1 + R} \tag{11}$$

**Table 1. Sodium binding coefficient R.**

| Combining effect theory | Expression for adsorption effect | Combined action coefficient |
|---|---|---|
| Linearcombination | $C_b = \alpha \times C_f$ | $R = \alpha$ |
| Langmuirisothermadsorptionbinding | $C_b = \frac{\alpha \times C_f}{1 + \beta C_f}$ | $R = \frac{\alpha}{(1 + \beta \times C_f)^2}$ |
| Freundlichisothermadsorptionbinding | $C_b = \alpha \times C_f^{\beta}$ | $R = \alpha \times \beta \times C_f^{\beta-1}$ |

$\alpha, \beta$—constant.

where $C_t$ represents the total content of sodium, $C_f$ represents the content of free sodium, and $C_b$ represents the content of bound sodium, $C_t = C_f + C_b$; $R$ represents the binding coefficient of sodium; $f_R$ is the coefficient of combined effect influence. The binding coefficients of chloride ions under different conditions are presented in Table 1. The variation pattern of $f_R$ with the binding effect coefficient is depicted in Fig 2b.

$$f_C = \exp(-3 + 1.2 \times CR) \tag{12}$$

$$f_J = \frac{4 \times J + 90}{2.9} \tag{13}$$

where $f_c$ is the molecular ratio influence coefficient; $f_J$ is the coefficient of influence on current density. The variation patterns of $f_J$ with current density and $f_c$ with molecular ratio are illustrated in Fig 2c and 2d.

The algorithm process is depicted in Fig 2.

## Development of a multifactor sodium diffusion prediction model for cathode carbon blocks

In order to facilitate the practical application of the microscopic numerical simulation method for sodium diffusion affected by multiple factors discussed in this article, it is essential to develop a predictive model for sodium diffusion within cathode carbon blocks.

Firstly, assuming that the sodium diffusion behavior within the cathode carbon block is one-dimensional and that the boundary concentration remains constant, the correction coefficient obtained by considering the aforementioned four factors is found to be linearly correlated with the effective diffusion coefficient of sodium. Consequently, a novel theoretical equation can be established:

$$\begin{cases} \frac{\partial C}{\partial t} = f_M \cdot \frac{\partial^2 C}{\partial x^2} \\ f_M = f_T f_R f_J f_C D_0 \end{cases} \tag{14}$$

where $f_M$ is the effective diffusion coefficient of sodium; $D_0$ is the initial sodium diffusion coefficient.

Based on Eq (14), it is evident that when the time for sodium to corrode the cathode carbon block reaches a certain period $t_{max}$, and other factors remain constant, the sodium diffusion coefficient hardly changes. At this juncture, discussing and studying the model becomes irrelevant. Therefore, focusing solely on the case where $t_0 < t < t_{max}$, solving Eq (14) provides a

sodium diffusion prediction model for cathode carbon blocks under multifactor correction:

$$
\begin{cases}
C(x, t, T, R, J, C) = C_0 + (C_s - C_0)\left[1 - \mathrm{erf}\left(\dfrac{x}{2\sqrt{f_T f_R f_J f_C D_0 t}}\right)\right] \\[2ex]
f_T = \exp\left[\dfrac{E_0}{R}\left(\dfrac{1}{T_{\mathrm{ref}}} - \dfrac{1}{T}\right)\right] \\[2ex]
f_R = \dfrac{1}{1 + R} \\[2ex]
f_J = \dfrac{4 \times J + 90}{2.9} \\[2ex]
f_C = \exp(-3 + 1.2 \times CR)
\end{cases}
\tag{15}
$$

From the revised model, it can be concluded that if the environmental conditions of the cathode carbon block remain unchanged, such as constant temperature and current density, the sodium concentration inside the cathode carbon block is solely dependent on the depth $x$ from the boundary and the duration of sodium erosion.

## Numerical simulation and result analysis

### Material parameters and numerical simulation

This study constructs a two-dimensional finite element model using a $\phi$0.05 m circular specimen identical to the one used in practical experiments, as depicted in Fig 3a. The material parameters for the finite element numerical simulation, as detailed in the text, are presented in Table 2. Initially, the concentration at the boundary nodes surrounding the model is initialized at 3%, while the concentration of all internal nodes is set to 0. The diffusion coefficient is obtained by multiplying the initial diffusion coefficient by correction coefficients that account for the influence of various factors. Following mesh division, the model is depicted in Fig 3b for transient calculation simulation purposes.

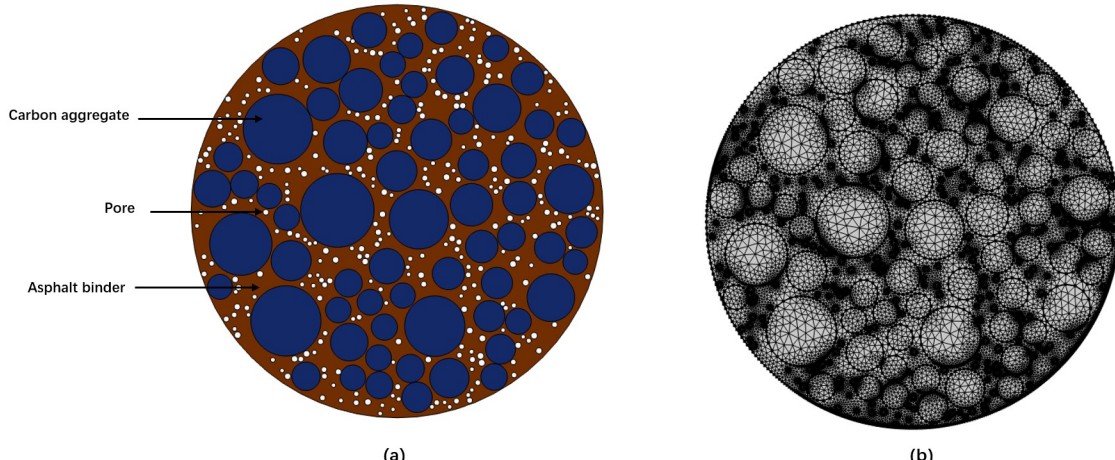

(a)                                                                (b)

**Fig 3. (a) Geometric model of sodium diffusion in carbon blocks and (b) finite element model of sodium diffusion in carbon blocks.**

**Table 2. Cathode carbon block material parameters [14].**

|  | Aggregate | Binder |
|---|---|---|
| *Density* | 1940kg/m$^3$ | 1200kg/m$^3$ |
| *Sodiumdiffusioncoefficient* | $8.9 \times 10^{-9}$m$^2$/s | $9.7 \times 10^{-8}$m$^2$/s |

The determination of the sodium diffusion coefficient in pores draws upon the analytical solution proposed by Zheng and Zhou [15] for the chloride ion diffusion coefficient in cement slurry, utilizing the effective medium theory. This approach establishes the expression for the sodium diffusion coefficient in relation to porosity:

$$D_\varphi = \frac{\alpha\varphi^{2.75}D_0}{\varphi^{1.75}(3-\varphi) + n(1-\varphi)^{2.75}} \tag{16}$$

where $\phi$ is the porosity of the carbon block; $N$ and $\alpha$ are undetermined parameters; $D_0$ is the initial sodium diffusion coefficient.

## Impact of pores on sodium diffusion

After studying 30 industrial carbon anodes in references [16, 17], the average total porosity was determined to be 16.2%, comprising 11.3% for kneading and calcination pores, 1.6% for aggregate cracks, 1.1% for green pores, and 2.2% for micropores.

In the present study, we focus on analyzing mixing pores and calcination pores, which constitute the highest proportion of pores. Numerical simulations were conducted under consistent conditions of aggregate morphology and content, with porosity levels set at 0%, 0.5%, 10%, and 15%, and a fixed calculation duration of 1000 seconds. Sodium concentration distribution maps for each porosity level are presented in Fig 4.

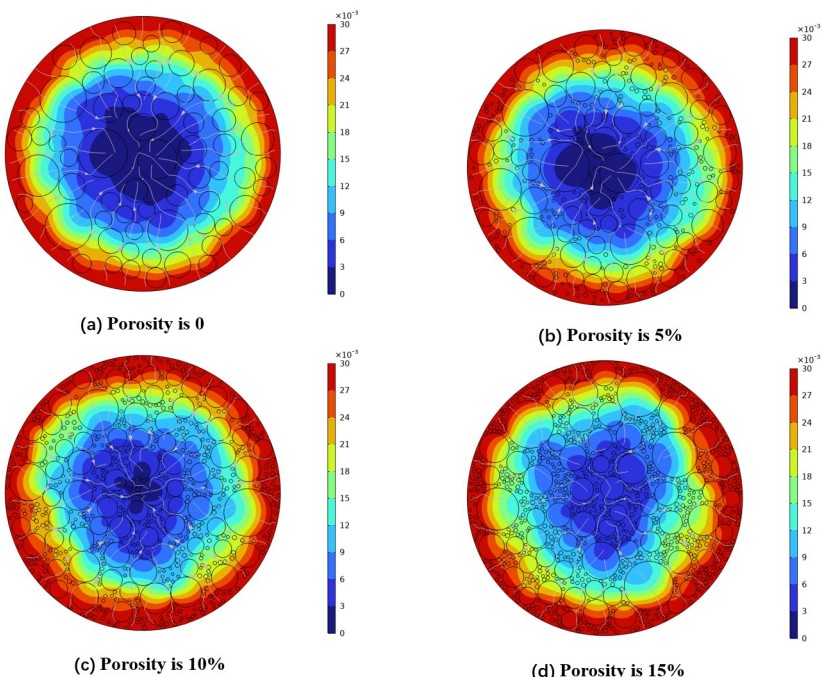

(a) Porosity is 0

(b) Porosity is 5%

(c) Porosity is 10%

(d) Porosity is 15%

**Fig 4. Nephogram of sodium concentration distribution with different porosity.**

It is evident from the figures that increasing porosity accelerates the diffusion rate of sodium, resulting in greater diffusion depth and a more complex diffusion path. This phenomenon can be attributed to the interconnected nature of pores at higher porosity levels, facilitating sodium permeation both axially and radially. This interconnectedness enables simultaneous permeation of sodium and electrolyte in multiple directions, thereby reducing diffusion resistance and enhancing the diffusion coefficient.

Under consistent conditions of aggregate morphology, content, and porosity, this study investigates the impact of electrolysis temperature, binding efficacy, molten phase molecular ratio, and cathode current density on sodium diffusion. A detailed analysis of the influence exerted by each of these factors is provided in the subsequent sections.

## Effect of electrolysis temperature on sodium diffusion

To investigate the impact of electrolysis temperature on sodium diffusion characteristics, a series of temperature values were selected: T = 940°C, 955°C, 970°C, 985°C, and 1000°C, with a calculation time of 1000 seconds. The sodium concentration distribution cloud maps under varying temperature conditions were obtained through numerical simulation, as depicted in Fig 5.

From the figure, it is evident that the diffusion path exhibits non-uniformity and anisotropic characteristics, and the sodium concentration contour lines show a distorted effect rather than a smooth curve. As ambient temperature increases, the diffusion rate of sodium accelerates significantly, leading to an increase in diffusion depth. The variation of sodium diffusion depth with electrolysis temperature is shown in Fig 6a.

As shown in the figure, at constant diffusion time, the diffusion depth of sodium increases with increasing ambient temperature. From 940°C to 1000°C, the extent of sodium diffusion increased by approximately 1.5 times across various diffusion intervals. This observation indicates that higher temperatures have a detrimental effect on enhancing the sodium corrosion resistance of carbon blocks.

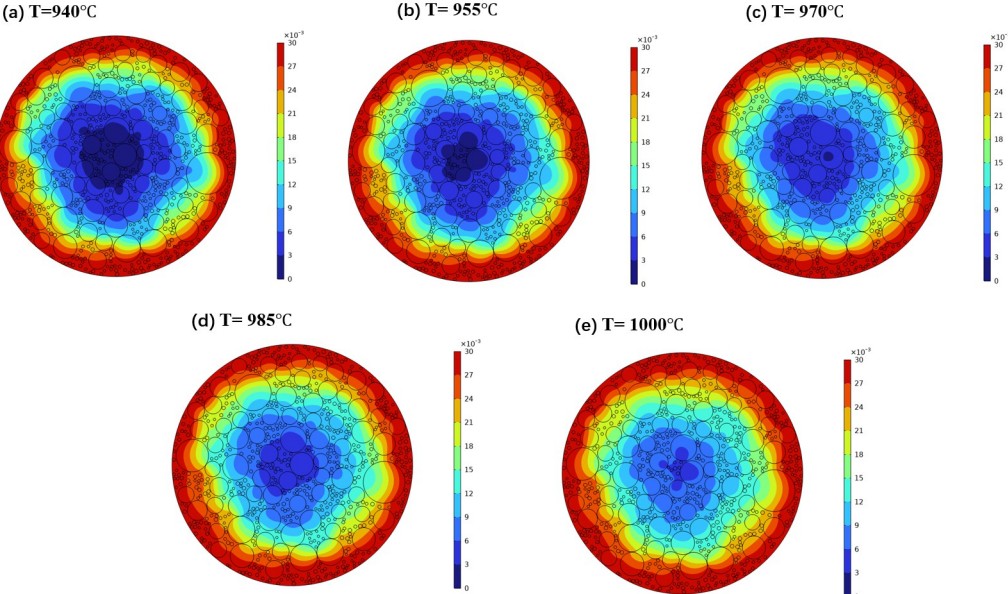

**Fig 5. Cloud map of sodium concentration distribution under different electrolysis temperature conditions.**

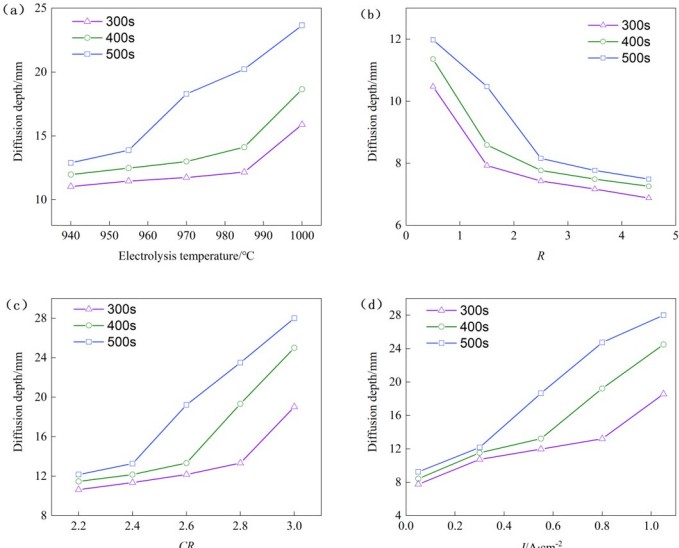

**Fig 6. The variation of sodium diffusion depth with (a) electrolysis temperature, (b) binding effect coefficient, (c) melt molecular ratio and (d) cathode current density.**

Fig 7a illustrates the correlation between sodium concentration and diffusion depth under varying temperature conditions. It is evident from the graph that at any given diffusion depth, sodium concentration increases with the electrolysis temperature. This suggests that higher temperatures have a catalytic effect on the sodium diffusion process. This phenomenon can be analyzed from two perspectives:

Firstly, with respect to sodium, increasing temperature enhances its activity, thereby accelerating its diffusion rate and promoting deeper penetration into the carbon block.

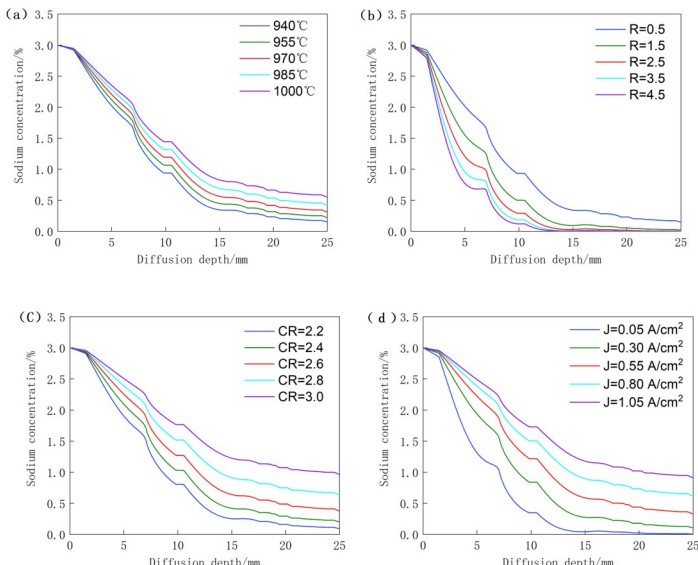

**Fig 7. The relationship between sodium concentration and diffusion depth under different factor conditions. (a)** Electrolysis temperature; (**b**) binding effect; (**c**) melt molecular ratio; (**d**) cathode current density.

Secondly, from the perspective of the carbon blocks, higher temperatures naturally expand their surface porosity, reducing their resistance to sodium corrosion and increasing the sodium concentration at equivalent depths.

## Influence of binding effect on sodium diffusion

To investigate the impact of sodium binding on the diffusion dynamics of sodium, a series of binding coefficients (R = 0.5, 1.5, 2.5, 3.5, and 4.5) were meticulously selected, with a consistent calculation period of 1200 seconds. Through detailed numerical simulations, sodium concentration distribution maps were generated, illustrating the diffusion characteristics under varying binding influences, as shown in Fig 8.

The visual data reveals a clear trend: as the binding coefficient increases, the rate of sodium diffusion significantly slows down, and the diffusion depth is substantially reduced. This trend is further corroborated by Fig 6b, which shows the progressive decline in sodium diffusion depth with increasing binding coefficients, while maintaining a constant diffusion duration. Notably, when the binding coefficient escalates from 0.5 to 8.5, the depth of sodium diffusion diminishes by approximately 40% across different diffusion intervals.

These findings highlight the efficacy of binding effects in enhancing the sodium corrosion resistance of carbon blocks, a development of considerable significance in the realm of materials science.

Fig 7b illustrates the correlation between sodium concentration and diffusion depth under various binding influences. It is evident from the graph that sodium concentration diminishes with increasing binding coefficients at any given diffusion depth. This phenomenon can be attributed to the inverse proportionality between the binding effect of sodium and the availability of free sodium within the pore structure. Additionally, sodium undergoes chemical reactions, leading to the formation of new substances that occupy the internal pores of the cathode carbon block. Consequently, this process impedes the diffusion rate of sodium, resulting in a descending trend in sodium concentration at equivalent depths.

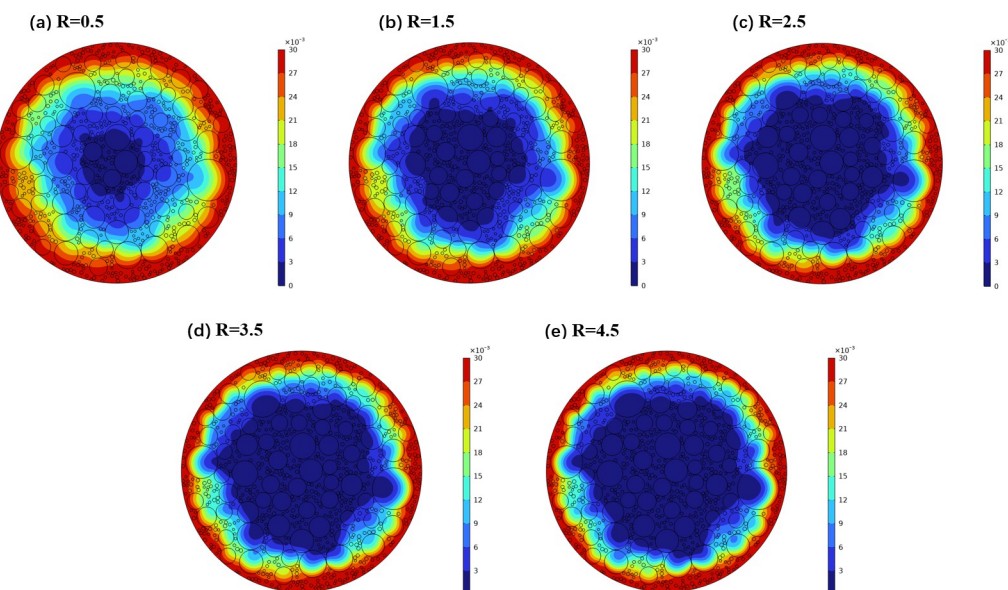

**Fig 8. Cloud map of sodium concentration distribution under different binding capacity conditions.**

## Effect of electrolyte molecule ratio on sodium diffusion

To intuitively reflect the influence of molecular ratio on sodium diffusion characteristics, five sets of molecular ratios were selected: CR = 2.2, 2.4, 2.6, 2.8, and 3.0. The calculation time was set at 1000 seconds. Through numerical simulation, sodium concentration distribution cloud maps under different molecular ratio conditions were obtained, as depicted in Fig 9.

The graph clearly shows that as the molecular ratio increases, sodium continuously diffuses into the interior of the carbon block, significantly increasing the diffusion depth. The variation of sodium diffusion depth with the ratio of molten molecules is shown in Fig 6c. As illustrated in the figure, when the diffusion time is constant, the depth of sodium diffusion also increases with the molecular ratio. When the molecular ratio increased from 2.2 to 3.0, the depth of sodium diffusion increased by about two times under different diffusion durations.

Based on the above analysis, selecting a smaller molecular ratio is beneficial for improving the sodium corrosion resistance of the carbon block, as it limits the diffusion of sodium into the interior. Previous researchers have also conducted extensive research on developing new electrolysis systems with low molecular ratios to address the significant energy and resource burden, as well as environmental challenges.

The correlation between sodium concentration and diffusion depth under varying molecular ratios is depicted in Fig 7c. It is evident from the graph that sodium concentration escalates with increasing molecular ratio at a constant diffusion depth, suggesting that an elevated molecular ratio facilitates the diffusion of sodium. Studies have demonstrated that an augmented molecular ratio within the melt intensifies the sodium ion concentration. This surge in sodium ion concentration can alter the constitution of the Stern double layer on the carbon cathode's surface, diminishing the thickness of the stationary layer and consequently reducing the dynamic electric potential $\xi$. The increased $\xi$ value enhances the molten metal's infiltration into the carbon cathode [18].

Moreover, as the molecular ratio of the molten liquid rises, the sodium fluoride content also increases. Considering the chemical reaction Al(I) + 3NaF(I) = 3Na(g) + AlF$_3$(I) instigated

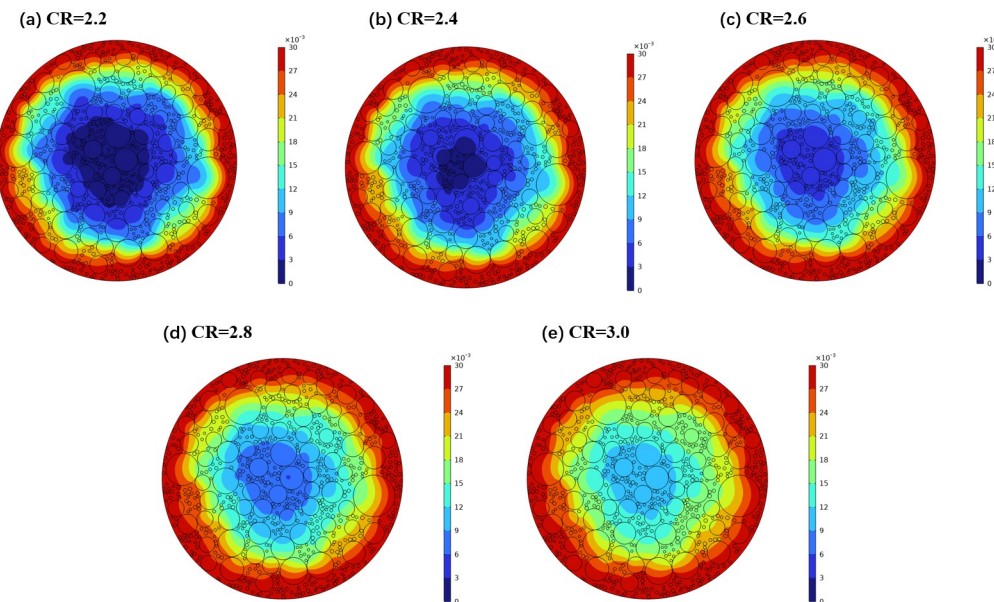

**Fig 9. Cloud map of sodium concentration distribution under different electrolyte molecular ratios.**

by sodium and the electrochemical reaction $Na^+ + e = Na(g)$, these two processes are more readily executed to the right, thereby increasing the yield of sodium produced and intensifying sodium penetration.

## Effect of cathode current density on sodium diffusion

To visually depict the impact of current density on the diffusion characteristics of sodium, various current densities (J = 0.05 A/cm$^2$, 0.30 A/cm$^2$, 0.55 A/cm$^2$, 0.80 A/cm$^2$, and 1.05 A/cm$^2$) were simulated for a duration of 1000 seconds. The sodium concentration distribution maps under these different current densities were obtained through numerical simulation, as illustrated in Fig 10. The figure shows that as current density increases, sodium continuously diffuses deeper into the carbon block, significantly enhancing the diffusion depth. The relationship between sodium diffusion depth and cathode current density is presented in Fig 6d. For a constant diffusion time, the depth of sodium diffusion also increases with the rise in current density. When the current density is elevated from 0.05 A/cm$^2$ to 1.05 A/cm$^2$, the depth of sodium diffusion increases by approximately 2.5 times across different diffusion durations. This observation suggests that an increase in current density is not beneficial for enhancing the sodium corrosion resistance of the carbon block.

The correlation between sodium concentration and diffusion depth under varying current conditions is depicted in Fig 7d. It is observed that at a given diffusion depth, the sodium concentration rises with the increase in current density, indicating that higher current density promotes the diffusion of sodium. This can be attributed to the fact that as the current intensity increases, more metallic sodium is produced by electrodeposition over the same period, leading to a greater amount of sodium permeation.

## Comparison of simulation and experimental analysis

To validate the reliability of numerical simulations, a comparative analysis is conducted with previous experimental results in reference [19]. Li et. al., employed a custom-made electrolytic

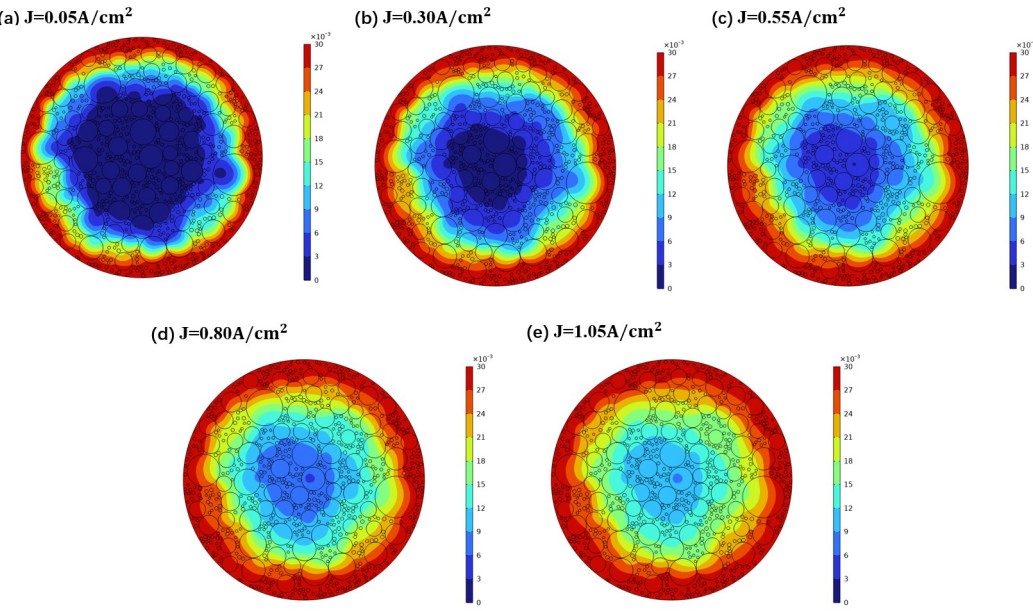

**Fig 10. Cloud map of sodium concentration distribution under different cathode current densities.**

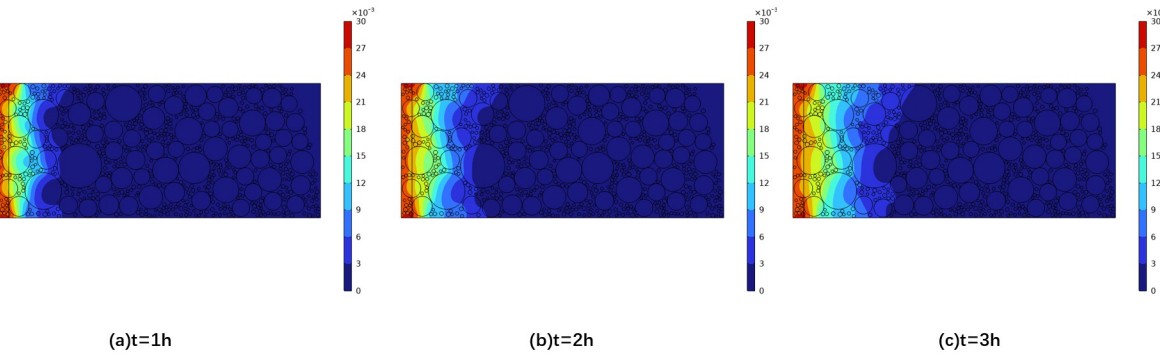

|        (a)t=1h        |        (b)t=2h        |        (c)t=3h        |

**Fig 11. Sodium concentration distribution cloud map.**

cell tube furnace to study sodium diffusion. The carbon block sample used was a cylindrical specimen with a diameter of 25 mm, immersed to a depth of 2 mm in the electrolyte. The electrolysis experiment was conducted at 960˚C and continuously protected by argon gas. The cathode current density was 0.5 A/cm$^2$, and the electrolyte molecular ratio was 2.5. The electrolysis time was 3 hours. Following the electrolysis process, the cathode specimen was sectioned axially for analysis using scanning electron microscopy (SEM-EDS) and phenolphthalein testing.

The parameters and boundary conditions of the simulation model were meticulously configured to align with the aforementioned experimental parameters. The sodium concentration distribution, as depicted in Fig 11. The 95% confidence interval for the experimental group is 0.9167 to 1.4141, while the 95% confidence interval for the simulation group ranges from 0.6443 to 1.0814. Given the overlap between these intervals, we may tentatively infer that the two data sets could be similar. However, to reach a more definitive conclusion, additional t-tests are required. The results indicated that the p-value exceeded the significance threshold, suggesting that there was no statistically significant difference between the two groups, thereby confirming the accuracy of the simulation. As time progresses, the depth of sodium diffusion intensifies, indirectly indicating the uneven diffusion process. A comparative analysis between the simulated and experimental values is presented in Fig 12, highlighting the consistent downward trend in sodium concentration with increasing diffusion depth. The alignment of the simulated and experimental values suggests a notable correlation, albeit with minor discrepancies at localized positions. These variations may be attributed to the three-dimensional finite element model's consideration of only the mixed and calcined pore types of the carbon block, potentially leading to differences in the sodium diffusion curve.

The above analysis indicates that numerical methods can accurately predict the sodium diffusion process in cathode carbon blocks, thus verifying the rationality and feasibility of the multi-factor sodium diffusion micro finite element model established in this study.

## Conclusion

To study the sodium permeation process and its influencing factors in cathode carbon blocks of aluminum electrolysis, this paper takes a microscopic approach, viewing the cathode carbon block as a composite heterogeneous material consisting of aggregates, asphalt binders, and pores.

1. A two-dimensional random aggregate digital simulation model was constructed to realistically depict the internal structure of the cathode carbon block. Simulation results indicate

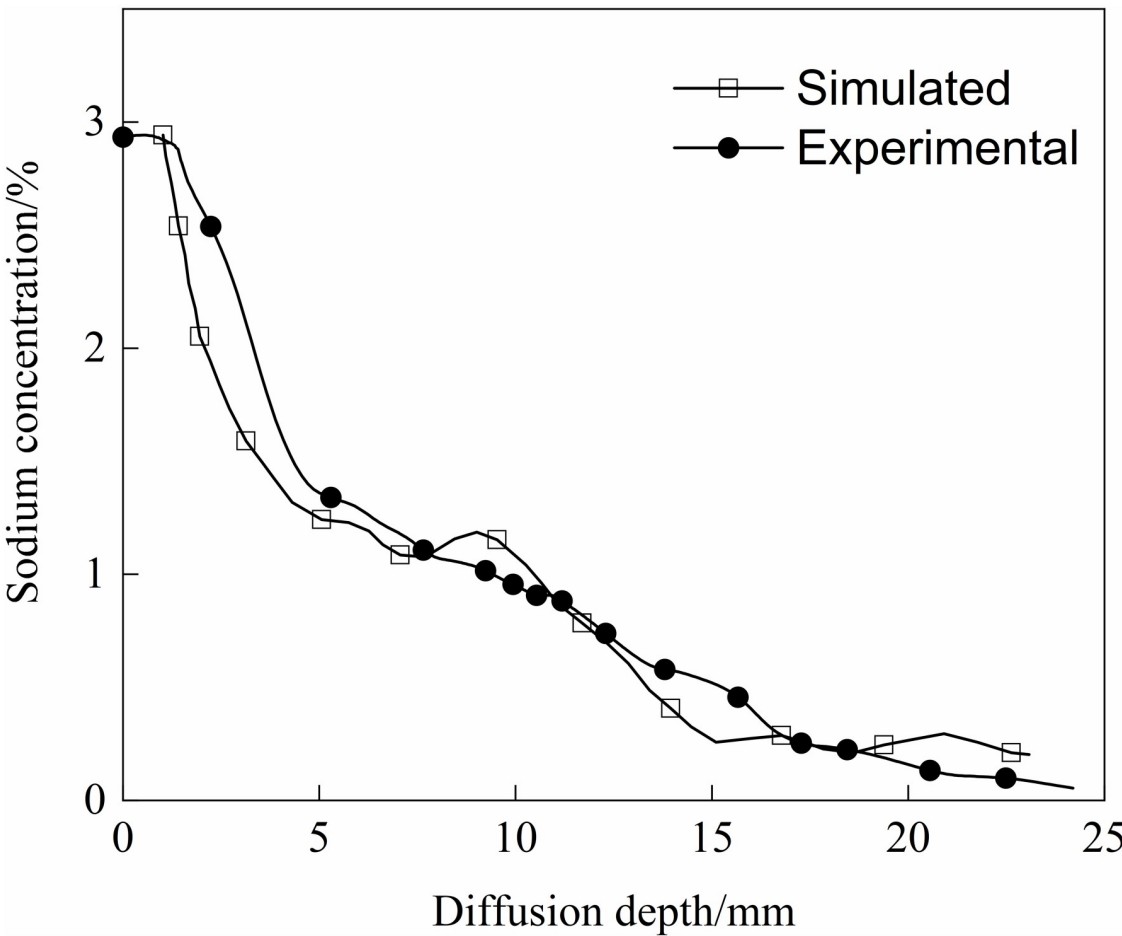

**Fig 12. Comparison of numerical simulation and actual test.**

that during sodium permeation, interconnected pores allow sodium and electrolyte to permeate in both axial and radial directions, resulting in simultaneous penetration in multiple directions. Higher porosity exacerbates sodium permeation.

2. The simulation further reveals that sodium permeation worsens with higher temperatures, reduced binding effects, increased current density, and higher molecular ratios. Comparisons with previous experimental results validate the feasibility of the simulation modeling approach proposed in this study.

3. The established sodium diffusion prediction model for cathode carbon blocks holds significant reference value for assessing the health status of aluminum electrolytic cells and analyzing the causes of cathode carbon block damage. Additionally, the study significantly impacts the production efficiency of the aluminum electrolysis industry, reduces costs, and promotes sustainable development. Future research will continue to explore this area, thereby providing new solutions for technological advancement and resource optimization within the industry.

It is important to note that while the key parameters of the cathode model in this study have been thoroughly defined within the computational framework, the two-dimensional mesoscale model has been simplified to a circular geometry. This simplification leads to macroscopic behaviors that may not fully align with the real-world system. To address this

limitation, the authors plan to introduce geometric randomness in future work, as well as develop a three-dimensional model. This approach will enable a better connection between the macroscopic characteristics of the material and its underlying microscopic properties, ultimately providing a more comprehensive understanding of the carbon cathode.

Carbon-based materials, as typical porous substances, are widely used in the electrode components of supercapacitors due to their fast ion adsorption and desorption properties [20], which facilitate efficient charge storage and rapid release. These materials not only exhibit low cost, excellent chemical stability, and high conductivity but also demonstrate superior electrochemical performance. Consequently, they play a crucial role in various energy storage devices, particularly in micro-supercapacitors (MSCs) [21]. Additionally, carbon-based materials are increasingly recognized as an ideal choice for wearable, flexible electronic systems because of their outstanding mechanical stability. Beyond supercapacitors, these materials have been successfully integrated with other conductive substances to form sulfur cathodes for lithium-sulfur (Li-S) batteries [22, 23], significantly enhancing battery performance. Furthermore, the advancement of carbon-based materials in other fields could accelerate the development of high-performance composite materials for use in aluminum reduction cell cathodes, which would inhibit sodium penetration, extend the lifespan of these cells, and slow the process of cathode degradation.

## Author Contributions

**Conceptualization:** Tianqi Xu.

**Data curation:** Huarong Qi.

**Methodology:** Tianqi Xu.

**Writing – original draft:** Chenglong Gong.

**Writing – review & editing:** Tianqi Xu, Yan Li.

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
