## [Decision Letter · Decision Letter 0]

6 Jan 2025

PONE-D-24-46130Numerical analysis of sodium diffusion in aluminum electrolysis cathode carbon blocks based on a microstructure multi-factor corrected modelPLOS ONE

Dear Dr. Xu,

Thank you for submitting your manuscript to PLOS ONE. After careful consideration, we feel that it has merit but does not fully meet PLOS ONE’s publication criteria as it currently stands. Therefore, we invite you to submit a revised version of the manuscript that addresses the points raised during the review process.

We look forward to receiving your revised manuscript.

Kind regards,

Sheng Yu

Academic Editor

PLOS ONE

Journal Requirements:

“This work was financially supported by National Natural Science Foundation of China (62062068) and Yunnan Province Young and Middle aged Academic and Technical Leaders Reserve Talent Project (202305AC160077).”

Reviewers' comments:

Reviewer's Responses to Questions

**Comments to the Author**

1. Is the manuscript technically sound, and do the data support the conclusions?

Reviewer #1: Yes

Reviewer #2: Yes

2. Has the statistical analysis been performed appropriately and rigorously? 

Reviewer #1: Yes

Reviewer #2: Yes

3. Have the authors made all data underlying the findings in their manuscript fully available?

Reviewer #1: Yes

Reviewer #2: Yes

4. Is the manuscript presented in an intelligible fashion and written in standard English?

Reviewer #1: Yes

Reviewer #2: Yes

5. Review Comments to the Author

Reviewer #1: This manuscript presents an interesting numerical investigation into the influence of various factors on sodium diffusion within cathode carbon blocks. The methodology is clearly explained and designed. The simulation results are both highly impressive and aligned with the experimental results, offering valuable insights for advancements in the aluminum electrolysis industry.

Key parameters are comprehensively established in the computational models. However, the two-dimensional mesoscale model is simplified as a circular structure. Increasing the randomness of the model’s geometry could enhance the reproducibility in the future. Additionally, a brief discussion on potential three-dimensional modeling could enhance the paper’s perspective and future studies. The literature review could be expanded to include more recent studies. It would be beneficial to use images with higher resolution to improve clarity.some recent work should be cited by this manuscript. 

1)Recent progress in electrode materials for micro-supercapacitors. *Iscience*.

2) MOF-Derived Nitrogen-Doped Porous Carbon Polyhedrons/Carbon Nanotubes Nanocomposite for High-Performance Lithium–Sulfur Batteries. *Nanomaterials*, *13*(17), 2416.

3) High-performance S cathode through a decoupled ion-transport mechanism. *Journal of Energy Storage*, *104*, 114588.

Overall, this work is well-structured and recommended for publication after minor revisions.

Reviewer #2: The study offers a predictive model to understand the sodium diffusion in cathode carbon blocks. Could you please provide some answers for the following questions:

1: Have you performed any statistical validation for the comparative results (e.g., error analysis, confidence intervals)?

2: Could you highlight its novel aspects more explicitly?

3: Mentioned in the introduction (Line 40-42). How can you method ensure that the experimental setup accurately reflects the real-world operating conditions?

There are several relevant works worth citing, including:

a. Lychee seed-derived microporous carbon for high-performance sodium-sulfur batteries. *Carbon*, 201, 864-870.

b. Recent advances in porous carbon materials as electrodes for supercapacitors. *Nanomaterials*, 13(11), 1744.

c. Three-dimensional nanostructured Co2VO4-decorated carbon nanotubes for sodium-ion battery anode materials. *Rare Metals*, 42(12), 4060-4069.

6. PLOS authors have the option to publish the peer review history of their article (what does this mean?). If published, this will include your full peer review and any attached files.

Reviewer #1: No

Reviewer #2: No

---

## [Author Response · Author response to Decision Letter 0]

10 Jan 2025

Dear editors and reviewers,

 We are grateful for your constructive comments and suggestions for our manuscript entitled “Numerical analysis of sodium diffusion in aluminum electrolysis cathode carbon blocks based on a microstructure multi-factor corrected model”(ID: PONE-D-24-46130).Your comments are very valuable and helpful for improving our manuscript. In the following, the responses to all the comments are provided one by one.

 We have tried our best to make all the revisions clear, and we hope that the revised manuscript can satisfy the requirements for publication.

 The main revisions in the new manuscript are:

1. Format Template for Articles have been updated.

2. The content requested by the reviewers have been added to the manuscript.

Sincerely,

Corresponding author.

Comments from the Editor

1.Please ensure that your manuscript meets PLOS ONE's style requirements, including those for file naming.

Author response: We have updated the manuscript format according to the template you provided. Specifically, it includes: 

1. Writing format for authors and institutions 

2. Image citation format 

3. Graphic file naming 

4. Added author contribution section 

5. Reference format

Specific changes can be found in the red highlighted section of the revised manuscript.

2. Please note that PLOS ONE has specific guidelines on code sharing for submissions in which author-generated code underpins the findings in the manuscript. In these cases, we expect all author-generated code to be made available without restrictions upon publication of the work.

Author response: We are very willing to provide the code required to build the geometric model in the manuscript, and have already shared the code in the manuscript in the database. We hope that the code we provide can better promote the development of this field. Please refer to DOI:10.6084/m9.figshare.28174739 for details

3. Thank you for stating the following financial disclosure:“This work was financially supported by National Natural Science Foundation of China (62062068) and Yunnan Province Young and Middle aged Academic and Technical Leaders Reserve Talent Project (202305AC160077).”

Author response: Thank you for pointing out the problem. We have supplemented the content of the fund in the manuscript as required and provided relevant explanations in the cover letter.

Author response: The content we submitted includes all the raw data required for replicating the research results, and we have shared our simulation data in the database as required, as detailed in DOI:10.6084/m9.figshare.281327. If any additional data is needed, we are more than happy to provide it.

Author response: We have rechecked the format of the references and made corrections where necessary. Specifically, as follows:

1.Deleted a reference：Li, B., Wang, J. Q., Gao, B. L.，et al. (2001). Study on penetration of carbon block in cathode of aluminum electrolysis. Light Metal, (07):37-40.

2. Added 4 references as requested by the reviewer：

Pan Z, Yu S, Wang L, Li C, Meng F, Wang N, Zhou S, Xiong Y, Wang Z, Wu Y, et al. Recent Advances in Porous Carbon Materials as Electrodes for Supercapacitors.[J]. Nanomaterials,2023,13:1744.

Xu Y, Yu S, Johnson HM, et al. Recent progress in electrode materials for micro-supercapacitors[J]. Iscience,2024.

Guo Y, Ying C, Ren L, et al. High-performance S cathode through a decoupled ion-transport mechanism[J]. Journal of Energy Storage,2024,104:114588.

Chen J, Yang Y, Yu S, et al. MOF-Derived Nitrogen-Doped Porous Carbon Polyhedrons/Carbon Nanotubes Nanocomposite for High-Performance Lithium–Sulfur Batteries[J]. Nanomaterials,2023,13(17):2416.

And we have marked it in the manuscript.

Comments from reviewers

Reviewer 1:

1. Key parameters are comprehensively established in the computational models. However, the two-dimensional mesoscale model is simplified as a circular structure. Increasing the randomness of the model’s geometry could enhance the reproducibility in the future.

Author response: We think this is a great suggestion. The two-dimensional mesoscale model in the manuscript has been simplified into a circular structure, and we plan to increase the randomness of the model geometry in the next article to improve future reproducibility. Please refer to the 411 to 416 lines of the manuscript conclusion for specific explanations.

2. Additionally, a brief discussion on potential three-dimensional modeling could enhance the paper’s perspective and future studies.

Author response: We draw inspiration from similar literature in other fields and further discuss the development direction of carbon cathodes. Please refer to lines 420-434 of the conclusion for details.

 3. The literature review could be expanded to include more recent studies. 

Author response: We sincerely appreciate your valuable feedback. We carefully reviewed the literature and added the latest research progress on sodium diffusion in cathode carbon blocks of aluminum electrolysis cells in the introduction section. Please refer to lines 42 to 47 of the introduction for specific modifications

4. It would be beneficial to use images with higher resolution to improve clarity. Some recent work should be cited by this manuscript. 

Author response: We appreciate your suggestion and apologize for our mistake. We have updated the resolution of the images in the latest manuscript to present our conclusions more clearly.

Reviewer 2:

1. Have you performed any statistical validation for the comparative results (e.g., error analysis, confidence intervals?

Author response: Thank you for your valuable feedback. We overlooked this point before. Now, we have added the data for this section, and the results show that there are overlapping confidence intervals between the comparison results. After conducting a t-test, it was found that there was no significant difference in the comparison results. Please refer to lines 370 to 377 of "Comparison of Simulation and Experimental Analysis" for the specific process.

2. Could you highlight its novel aspects more explicitly?

Author response: We are pleased to highlight the originality of this manuscript. Currently, the simulation methods for sodium diffusion primarily include homogenization theory and microstructure models. However, many existing microstructure models fail to adequately account for the impact of cathode carbon block porosity on sodium diffusion. In reality, the carbon cathode materials used in aluminum electrolysis cells typically have a porosity ranging from 15% to 30%. Yet, most microscale models of cathode carbon blocks overlook the influence of these pores on sodium diffusion, which does not fully align with actual conditions. In this manuscript, we emphasize the significant role of pores as a key factor affecting sodium diffusion, and integrate this consideration with environmental factors to establish a comprehensive model for sodium diffusion in carbon cathodes.

3. Mentioned in the introduction (Line 40-42). How can you method ensure that the experimental setup accurately reflects the real-world operating conditions?

Author response: We greatly appreciate the question you raised. The sodium diffusion process in cathode carbon blocks of aluminum electrolytic cells is both prolonged and intricate. In practical applications, numerical simulation technology offers significant advantages in terms of time efficiency, accuracy, and convenience, particularly given the long duration of macroscopic testing. Therefore, utilizing numerical simulations to investigate sodium diffusion behavior can substantially enhance both the efficiency and accuracy of research efforts. In the comparison between the manuscript's simulation and experimental analysis (lines 359 to 367), we emphasized that the simulation's boundary conditions should closely approximate the actual operating conditions. However, it is important to note that the results from finite element simulations cannot fully replicate those of real-world operations due to various environmental factors that introduce errors. Despite these limitations, finite element simulations continue to be invaluable in addressing practical problems.

4. There are several relevant works worth citing.

Author response: We believe that the literature you recommended is of great reference value to our research, and we have cited them as our references in the manuscript.

---

## [Editor Report · Decision Letter 1]

13 Jan 2025

Numerical analysis of sodium diffusion in aluminum electrolysis cathode carbon blocks based on a microstructure multi-factor corrected model

PONE-D-24-46130R1

Dear Dr. Xu

We’re pleased to inform you that your manuscript has been judged scientifically suitable for publication and will be formally accepted for publication once it meets all outstanding technical requirements.

Kind regards,

Sheng Yu

Academic Editor

PLOS ONE

---

## [Editor Report · Acceptance letter]

16 Jan 2025

PONE-D-24-46130R1 

PLOS ONE

Dear Dr. Xu, 

I'm pleased to inform you that your manuscript has been deemed suitable for publication in PLOS ONE. Congratulations! Your manuscript is now being handed over to our production team.

Kind regards, 

on behalf of

Dr. Sheng Yu 

Academic Editor

PLOS ONE